# Genetic Approaches to Increase Arabinoxylan and β-Glucan Content in Wheat

**DOI:** 10.3390/plants12183216

**Published:** 2023-09-08

**Authors:** Anneke Prins, Ondrej Kosik

**Affiliations:** 1Department of Sustainable Soils and Crops, Rothamsted Research, Harpenden AL5 2JQ, UK; anneke.prins@rothamsted.ac.uk; 2Department of Plant Sciences for the Bioeconomy, Rothamsted Research, Harpenden AL5 2JQ, UK

**Keywords:** wheat, dietary fiber, arabinoxylan, β-glucan, breeding, QTL, marker-assisted selection, wide cross

## Abstract

Wheat is one of the three staple crops feeding the world. The demand for wheat is ever increasing as a relatively good source of protein, energy, nutrients, and dietary fiber (DF) when consumed as wholemeal. Arabinoxylan and β-glucan are the major hemicelluloses in the cell walls and dietary fiber in wheat grains. The amount and structure of DF varies between grain tissues. Reducing post-prandial glycemic response as well as intestinal transit time and contribution to increased fecal bulk are only a few benefits of DF consumption. Dietary fiber is fermented in the colon and stimulates growth of beneficial bacteria producing SCFA, considered responsible for a wide range of health benefits, including reducing the risk of heart disease and colon cancer. The recommended daily intake of 25–30 g is met by only few individuals. Cereals cover nearly 40% of fiber in the Western diet. Therefore, wheat is a good target for improving dietary fiber content, as it would increase the fiber intake and simultaneously impact the health of many people. This review reflects the current status of the research on genetics of the two major dietary fiber components, as well as breeding approaches used to improve their quantity and quality in wheat grain.

## 1. Introduction

Dietary fiber is defined by the CODEX Alimentarius Commission as carbohydrate polymers that are neither hydrolyzed nor absorbed in the small intestine and have a degree of polymerization of at least three monomeric units [1]. Most of these carbohydrates are found in plant cell walls where they provide structural support, but they also have other functions, such as regulation of growth and signaling during plant development and stress [2,3,4].

Dietary fiber has many beneficial effects on human health, including improving cardiovascular health, regulating blood lipids, modulating post-prandial blood glucose, modulating the amount and diversity of gut microbiota, and decreasing intestinal inflammation [5,6,7,8,9]. It may even mitigate loss of skeletal muscle mass in older adults [10]. Internationally, regulatory bodies allow specific health claims to be made for food products that contain a certain minimum amount of dietary fiber (EFSA Panel on Dietetic Products, Nutrition and Allergies, 2010; USFDA, 2018). 

The health benefits of dietary fibers relate to their different physicochemical characteristics (solubility, viscosity, and fermentability). Non-viscous and non-fermentable dietary fiber (such as wheat bran) plays a role in promoting gut motility largely through its bulking characteristics [8,11]. Viscous dietary fiber (mostly soluble dietary fiber) is related to cardiovascular and glycemic benefits through its effect on the rheological properties of the digesta, which decreases the rate of digestion and absorption of macronutrients [12,13,14]. Fermentable dietary fiber acts as a prebiotic, providing a carbon source for the growth of beneficial microbiota in the colon. Fermentation of these dietary fibers results in specific changes in the composition and/or activity of gut microbiota, which supports host health [15,16,17]. Prebiotic activities of cereal grain polysaccharides have been thoroughly tested. In vitro fermentation studies have shown the possibility of synergistic activities between arabinoxylan (AX) and β-glucan promoting the increase in the total number of bacteria as well as beneficial groups of *Bifidobacterium* and *Clostridium coccoides*/*Eubacterium* groups. Similarly, the concentration of short-chain fatty acids (SCFAs) increased. Larger amounts of AX appear to prompt the production of acetate, whereas β-glucan promotes the production of propionate [18]. 

Consumption of dietary fiber varies between countries, gender, and age. When considering Europe and North America only, adult males consume 15–25 g per day, while adult females consume 14–21 g per day. In general, consumption is higher in Europe than in North America [19]. The recommended daily intake of dietary fiber for improved health is approximately 30 g for adults [20] with a specific recommendation of 10 g AX per day and 3 g of β-glucan per day to obtain specific health effects [19]. Only a minority of countries reach the very minimum recommended intake of dietary fiber per day (25 g; Norway, Germany, and Hungary), and this is only achieved in the male segment of the population [21]. Grain products are the largest source of dietary fiber in Europe and North America (32–49%), with the majority of this coming from wheat and bread alone, contributing approximately 20% of the average daily intake of dietary fiber [19]. In 2020, approximately 71% of wheat produced globally was consumed as food (FAO.org, Figure 1). This is predicted to remain stable as a percentage of total food consumption (~70%), while the absolute consumption as food is predicted to increase by 57 Mt by 2031 (FAO.org; OECD/FAO, 2022). Therefore, wheat is a good target for the improvement of dietary fiber content, as it would impact the health of people worldwide.

### Wheat Grain and Dietary Fiber

Wheat grain consists of three major milling fractions: the bran, starchy endosperm, and germ. These fractions consist of different tissue types which are characterized by distinct chemical compositions. 

The bran fraction contains several differentiated layers, namely the pericarp (outer and inner), the seed coat, and nucellar epidermis (hyaline layer). This fraction has a high cellulose, AX, and phenolic content compared to the starchy endosperm [22]. The outer cell layers enclose the aleurone and starchy endosperm, which constitutes the largest part of the grain. The starchy endosperm contains a large amount of glucose in the form of starch, with less AX and phenolic compounds compared to outer layers, while the aleurone contains a greater amount of protein and lipids [23]. The germ consists of the embryonic axis and scutellum, and also contains more protein and lipids than fractions in the center of the grain [23]. In general, the milling process separates the different layers of the grain to separate flour and semolina from the bran and germ [22]. The endosperm (consisting of the aleurone layer and starchy endosperm) constitutes the largest proportion of the grain (approximately 83%), while the germ represents about 3% and the peripheral layers approximately 14% [22]. While the aleurone is part of the endosperm, it is removed during the milling process and is considered part of the bran by millers [24,25]. The aleurone layer is typically only one cell layer thick and represents the only live endosperm tissue at maturity. It does, however, represent a rich source of nutrients including dietary fiber, proteins, minerals, vitamins, and flavonoids [25,26,27].

In mature wheat grain, about 12% accounts for cell wall (CW) polysaccharides. The major components in the CW and the major dietary fiber components in wheat grain are AX (accounting for 65–70% of CW material) and β-glucan (around 25%) with small amounts of cellulose (2–3%) and glucomannans (2–7%). Additionally, two non-CW-derived, non-starch polysaccharides contributing to dietary fiber in wheat endosperm are water-soluble arabinogalactan protein (AGP), accounting for 0.4% dry weight (DW), and fructans, accounting for ~1.5% DW of white flour [28,29,30]. Finally, resistant starch, the non-digestible portion of starch reaching the colon intact accounting for 0.8% of endosperm DW, can also be included as wheat dietary fiber [31].

Due to the wide-ranging health benefits of dietary fiber, wheat is a prime target for nutritional improvement through the increase in constituents like AX and β-glucan. Below, we review the current status of research on the genetics that underpin the two major dietary fiber components, as well as breeding approaches used to improve the content and quality of AX and β-glucan in wheat. 

## 2. Arabinoxylan (AX)

Arabinoxylan (AX) is thought to play a key role in regulating the strength of wheat endosperm primary cell wall, where it is the most prevalent hemicellulose. The AX backbone is made up of β-1,4-linked xylopyranose units decorated with α-arabinofuranose linked to O-2 and/or O-3 carbon of xylose units. The degree of arabinosyl substitution differs between cells within the endosperm [28].

Some of the 3-linked arabinose residues can be further decorated with ferulic acid at 5-O-position. Ferulic acid decorates only 0.2–0.4% (*w*/*w*) of WE-AX and 0.6–0.9% (*w*/*w*) of WU-AX in starchy endosperm [32]. Oxidation of ferulates and their close proximity on two adjacent AX chains can form diferulate crosslinks, which affect the physicochemical properties of AX in the cell wall. Particularly, this affects AX solubility and viscosity of water extract [30]. The majority of hydroxycinnamic acid derivates (ferulic, coumaric, and sinapic acids) and their dimers were found in aleurone and scutellum. Ester-linked p-coumaric acid (pCA) and ferulate dimers (8,5′, 8–5′ benzo and 5–5′ diferulates) were not detected in pure dissected wheat starchy endosperm [22]. Members of the “Mitchell clade” within the BAHD acyltransferase superfamily are involved in xylan esterification [33,34] but their mode of action has not been fully established yet.

A more complex form of AX occurs in outer layers of the wheat grain, in the pericarp and testa, where it can be also substituted with glucuronic acid [35].

The total arabinoxylan (TO-AX) content of wheat grain comprises water-extractable (WE-AX) and water-unextractable arabinoxylan (WU-AX) fractions. Other than water solubility, WE-AX and WU-AX differ in the polymer chain length, the arabinose:xylose ratio, and degree of feruloylation, resulting in different physical properties and biological functions. Whereas WU-AX has a high molecular weight, feruloylation, and a low degree of arabinose substitution, WE-AX has a higher degree of arabinosylation and lower polymer chain length [36]. 

The genotype (G) and the environmental conditions (E) determine the properties and content of arabinoxylan. Durum wheat, compared to winter and spring bread wheat, has in general lower variability in the AX properties. The AX content in wheat and other main cereals is summarized in Table 1.

### 2.1. Genetic Control of Arabinoxylan Content and Structure

Identification of candidate genes involved in AX biosynthesis has been mainly focused on hexaploid wheat. First, using an approach involving the differential expression of orthologues genes, several transglycosylases (GTs) have been identified. This included GTs from families GT2 *CslC*, GT43, GT47, GT48, GT61, GT64, and GT77. Later, the number of candidates involved in AX synthesis was reduced down to three GT families, GT43, GT47, and GT61. Additionally, genes from the BAHD acyl-CoA transferase superfamily were identified as responsible for the feruloylation of AX [33].

Transcriptome analysis of developing wheat starchy endosperm using RNA-Seq identified a number of homologous genes from the above-mentioned GT families involved in many things, whether in xylan backbone synthesis (GT43 and GT47) or the arabinosylation of xylan (GT61). Their expression profiles vary to a large extent during the grain filling stage between 10 and 28 dpa, respectively. Wheat *TaGT47_2* and *TaGT61_1* are the two most highly expressed AX biosynthesis genes in the endosperm during this grain developmental stage. Only four GT43 genes were expressed in starchy endosperm, *TaGT43_1* and *TaGT43_2* in high abundance and *TaGT43_3* and *TaGT43_4* in low abundance. Besides the highly abundant *TaGT47_2*, another eight GT47 genes, namely *TaGT47_1*, *TaGT47_3*, *TaGT47_4*, *TaGT47_5*, *TaGT47_6*, *TaGT47_10*, *TaGT47_13,* and *TaGT47_14,* were found at low abundance. Finally, another five GT61 genes, with abundance substantially lower than the highly expressed *TaGT61_1*, were identified in developing endosperm, namely *TaGT61_2*, *TaGT61_9*, *TaGT61_11*, *TaGT61_13,* and *TaGT61_14*. Additionally, five BAHD genes identified as candidates for the feruloylation of AX were found in wheat developing grain, *TaBAHD1* to *TaBAHD5*, with much greater expression levels in whole grain compared to starchy endosperm only [50]. This correlates with a greater level of feruloylation in the aleurone.

The backbone of AX is considered to be synthesized by xylan synthase, a complex of three subunits encoded by IRX14, IRX9 (both GT43 family), and IRX10 (GT47). Whereas IRX10 is believed to be responsible for the catalysis of xylan synthesis, IRX9 and IRX14 are required for positioning the xylan synthase complex in the Golgi [51].

Utilizing various transgenic wheat lines with RNAi constructs targeting suppression of AX synthetic genes led to alterations in quantity and structure of AX in wheat endosperm. Targeting the most highly expressed genes in xylan backbone synthesis *TaGT43_1*, *TaGT43_2* (IRX9), and/or *TaGT47_2* (IRX10) in a single RNAi construct resulted in a 40–50% decrease in AX amount and a decrease in AX polysaccharide chain length in TaGT43_2 RNAi [52,53]. Similarly, triple RNAi transgenic lines for GT43_1, GT43_2, and GT47_2 genes and for a triple knock-out mutant of TaIRX9b, stacking loss-of-function alleles in A,B,D homologues of GT43_2, resulted in a similar reduction in total AX of 47% and 65%, respectively [53,54].

The existence of multiple copies of xylan backbone synthesis genes present in the hexaploid wheat genome led to a hypothesis for the presence of more than one xylan synthase complex, e.g., one composed of highly expressed genes, GT43_1 (IRX14), GT43_2 (IRX9), and GT47_2 (IRX10), and another composed of subunits encoded by less expressed IRX9 (GT43_6, GT43_3, or GT43_4), IRX14 (GT43_10), and IRX10 (GT47_1 or GT47_4) genes. The existence of various xylan synthase complexes would explain the synthesis of different forms of AX and the necessity for a minimum AX level in wheat grain to maintain normal seed development [53,55]. This hypothesis still remains to be confirmed. On the other hand, a xylan synthase complex has been described in asparagus, where the co-expression of *AoIRX9*, *AoIRX10,* and *AoIRX14A* is required to form this complex and an equivalent formed of IRX9L, IRX10L, and IRX14 has been argued to exist in *Arabidopsis thaliana* [51,56].

Two GT61 genes, *TaXAT1* and *TaXAT2,* abundantly expressed in wheat grain, are involved in AX biosynthesis by decorating xylan backbone. RNAi suppression of *TaXAT1* showed a 70–80% decrease in mono-substituted AX oligosaccharides, indicating that TaXAT1 is responsible for the majority of α-1,3-linked arabinofuranosyl mono-substitution of wheat AX. Similarly, heterologously expressed *TaXAT2* in *Arabidopsis gux*-background lacking the glucuronic acid decoration on the xylan backbone confirmed its role in adding 3-linked arabinosylation to the xylan backbone. There was no effect on di-substituted AX oligosaccharides found in any of these constructs [57,58]. Genes involved in the biosynthesis of AX are summarized in Table 2 and their position on wheat chromosomes is depicted in Figure 2A.

### 2.2. QTLs Linked to Arabinoxylan Content

One of the most important steps in crop improvement is to detect and precisely localize chromosomal regions (loci) underlying agronomically relevant quantitative traits—in our case, arabinoxylan content, its solubility and structure, or in general, grain fiber content.

Quraishi et al. used 156 wheat lines (*Triticum aestivum*) from the HEALTHGRAIN diversity panel [64] and identified 12 QTLs for grain dietary fiber in bread wheat (*T. aestivum*) on chromosomes 1B, 3A, 3D, 5B, 6B, 7A, and 7B and three meta-QTL for WE-AX-related viscosity on chromosomes 1B, 3D, and 6B. They also demonstrated that the major locus for this trait is located on chromosome 1B. They identified 73 candidate genes for being involved in grain fiber content [65].

In hexaploid wheat cultivar (Berkut × Krichauff doubled haploid cross, grown at two sites in Australia in 2007 and 2009), Nguyen et al. reported QTLs for total AX content on chromosomes 1A, 2A, 3D, 4D, 6B, and 7A, where two of these QTLs were found to have a major effect (on chromosomes 2A and 4D) on grain AX. These QTLs (on 2A and 4D) were further validated and all lines carrying both favorable alleles contributed significantly to an increase in grain AX [66].

Marcotuli et al. investigated the genetic variability of AX content in tetraploid wheat genotypes (panel of 104 *Triticum turgidum* wheats grown in southern Italy in summer 2012) and identified genetic regions attributable to grain AX content characterized by SNP markers using a genome-wide association study (GWAS). In total, 37 significant marker-traits (MTAs) identifying 19 quantitative trait loci (QTLs) associated with AX content were revealed. Nine out of these markers showed high sequence similarity with annotated genes encoding enzymes implicated in AX biosynthesis, including glycosyltransferases from the GT1, GT31, and GT48 families as well as glycosyl hydrolases from families GH9, GH35, and GH47 [67].

Yang et al. analyzed 240 recombinant inbred wheat lines derived from PH82–2 (hard winter wheat) × Neixiang 188 (soft winter wheat) cross for wholemeal total (TO-AX) and water-unextractable (WU-AX) and water-extractable AX (WE-AX) content. They identified four additive QTLs for TO-AX content (on chromosomes 1B, 1D, 3B, and 5B), two additive QTLs for WU-AX content (on chromosomes 1B and 1D), and nine QTLs for WE-AX content (on chromosomes 1A, 1B, 2B, 3B, 5A, 5B, 6B, 7A, and 7B), explaining up to 14.6%, 2.4%, and 1.4%, respectively, of phenotypic variance [68].

Most of the research has been conducted on wholemeal flour, except for the two most recent works that have been carried out on refined/white flour.

Analyzing crosses between the high-AX cultivar of Chinese spring wheat Yumai-34 and three cultivars with an average amount of AX (Ukrainka, Altigo, and Claire) identified several QTLs, including a major QTL on chromosome 1B for high relative viscosity/total AX. KASP (Kompetitive allele specific PCR) marker for 1B QTL, the Yumai-34 high-AX allele, was validated, while analyzing a Yumai-34 cross with another high-AX variety (Valoris) identified a second major QTL on chromosome 6B for relative viscosity, with Valoris being the increasing allele [69].

Ibba et al. used a set of 175 bread wheats to identify significant genomic regions associated with AX content in white flour. They found two QTLs associated with total AX and seven with WE-AX content variation. The region on chromosome 1B coincides for both AX fractions and was the most significantly associated one with the observed phenotypic variation. Four KASP markers for the 1B QTL were developed and validated, and a candidate gene encoding glycosyl transferase GT61 [57] associated with the observed variation in AX was also identified [70].

The identification of QTLs and KASP markers for AX-related traits will significantly contribute to the improvement of wheat grain dietary fiber in breeding programs. All identified QTLs for AX-related traits are summarized in Table 3.

### 2.3. Breeding Approaches to Improve Arabinoxylan

To our knowledge, increasing dietary fiber, and AX particularly, has not been a goal of any breeding program so far. There is evidence in the scientific literature that it is possible to increase arabinoxylan content and both water-extractable and unextractable AX in white flour without compromising grain quality or yield [71]. Several mapping populations have been developed and used to determine the genetic control of AX. General breeding approaches to improve AX are summarized in Figure 3.

The first mapping populations were developed in the 1990s, by Van Deynze and colleagues in 1995 and by a group at INRA Clermont-Ferrand in 1997. The first population, where 115 lines were derived from a cross between synthetic wheat (a cross between *T. tauschii* and Altar 84—durum wheat cultivar) and Mexican spring wheat Opata 85, was used to map group 1 chromosomes of *Triticeae* species [72]. In the second population, 106 intervarietal doubled haploid lines obtained from a cross between French cultivar Courtot and Chinese Spring were analyzed for establishing a molecular map using restriction fragment length polymorphism probe (RFLP) [73]. These two mapping populations of bread wheats were used to analyze water-extractable AX, relative viscosity of wheat flour aqueous extract, and arabinose-xylose (Ara/Xyl) ratio. A QTL for relative viscosity and Ara/Xyl ratio was found on chromosome 1BL explaining 32–37% of the variation in relative viscosity and 35–42% variation in Ara/Xyl ratio in wheat endosperm [74].

Two recombinant populations derived from crosses between high- and low-WE-AX parents (Valoris × Isengrain and RE0006 × CF0007) were used together with the measurement of relative viscosity of flour extract as an indicator of WE-AX content. This allowed Charmet et al. to identify a QTL on chromosome 6B present in both populations to explain up to 59% of the phenotypic variation for WE-AX content and viscosity [75].

Another attempt to improve the dietary fiber content in white flour has been made by crossing Yumai-34, a Chinese wheat cultivar high in AX content (released in 1988), and three Central European wheat varieties (Lupus, Mv-Mambo, Ukrainka; Table 4) well-adapted to environmental conditions, including high productivity and good abiotic stress resistance. In total, 31 agronomically attractive lines, combining high total (TO-AX) and water-extractable (WE-AX) AX in flour, were selected. The increase in WE-AX content was greatest in the genetic background of the Ukrainka variety, and the TO-AX content was significantly higher in five Yumai34 × Ukrainka and five Yumai34 × Lupus lines than in the Yumai34 parent [71].

The addition of *Aegilops* chromosomes improved TO-AX content significantly only after adding U chromosomes 5U^g^ and 7U^g^ of *Ae. geniculata* and 1U^b^ of *Ae. biuncialis* to Chinese Spring bread wheat cultivar (Table 4). Water-extractable AX (WE-AX) was significantly improved after adding *Aegilops* U and M chromosomes 3U^g^, 4U^g^, 5U^g^, 6U^g^, 7U^g^, 5M^g^, and 7M^g^ of *Ae. geniculata* and 2M^b^ and 7M^b^ of *Ae. biuncialis* to bread wheat under both optimal and drought conditions [76].

### 2.4. Plant Breeding and Arabinoxylan Heritability

There is a substantial variation in components of wheat grain, including dietary fiber and its quantity and composition. These variations are consequences of three effects: genetic differences between lines (genotypes), influence of environmental conditions (including weather—hot, dry, wet, etc., agricultural practices, and soil conditions), and interactions between these two—genotype and environment (G × E) interactions. The “broad sense heritability” of dietary fiber and other grain components can be calculated by comparing the sample composition of multiple genotypes grown in multiple environments (sites and/or years) [79]. The availability of data is the limiting step in estimating heritability for a given trait and assessing whether or not this is available to breeders.

A dataset of 26 wheat lines grown in four locations (in a single year) in six environments (over three years), to simulate the wide range of climatic conditions within EU member states, was used in statistical models to find out the relationship between genotype and environment effect (G × E interaction) on fiber content in wholemeal bran and flour fractions. For dietary fiber-related traits, the following model was used: X = µ + E + G + G × E + ε, where µ is the grant mean, E the environment main effect, G the genotype main effect, G × E the interaction between the two main effects, and ε the residual error. The individual dietary fiber traits showed differences to which they vary between lines and environment, with the highest variability for water-extractable AX (WE-AX) in flour [38,80]. The dietary fiber traits, in bran and flour fractions, show high genetic heritability. Notably in flour, total (TO-) and water-extractable arabinoxylan (WE-AX) showed heritability of about 70% and 60%, respectively [79]. For TO-AX and WE-AX, the heritability was lower, 32% and 47%, respectively, with great variation attributed to the environmental effect, 30% for TO-AX and 39% for WE-AX in bran [38].

Genetic improvement has played a pivotal role in improving the yield and performance of wheat in the post-war period. Authors in [81], using nearly 53,000 observations, calculated that in the UK between 1982 and 2007, up to 88% of yield improvement in cereals can be attributed to genetic improvement rather than to changes in agronomy. Previously, between 1948 and 1981, genetic and environmental effects were of roughly equal importance, but plant breeding of winter wheat still contributed to around 60% of improvement. This includes the introduction of dwarfing *Rht*-genes in the 1970s (the “green revolution”), which increased the harvest index and the yield [82]. Within all these years (1948 to 2007), the grain yield of winter wheats increased from around 5 to 8 t/ha.

The effect of intensive breeding on starch and protein, mainly gluten, is widely documented. The knowledge of the impact of intensive breeding on other bioactive components is sparse, including arabinoxylan.

A study comparing the AX content in a small panel of old and modern Italian durum wheats showed no difference in AX and β-glucan content in wholemeal and semolina (refined flour of durum wheat) but showed higher solubility of AX in modern varieties [83]. 

Another study, performed by [82], analyzed 39 UK-adapted wheat cultivars from years between 1790 and 2012 (‘UK Heritage Wheats’) grown in a randomized 3-year field trial experiment, analyzing arabinoxylan and β-glucan (dietary fiber), soluble sugars, and polar metabolites. The study indicated a strong effect of environment on these traits but also concluded an increasing trend in amounts of AX accounting for 21% of the total variations (although not a main breeding trait) due to the effect of intensive breeding [82].

A comparison of total AX content in white flours from ~150 wheat genotypes showed wide variation between 1.4 and 2.8% DW with about 25–50% of the total being water-soluble. About 70% of the total AX and 60% of the water-soluble AX variation can be attributed to genotype [84,85].

The contents of total (TO-) and water-extractable (WE-) pentosans (measured as a proxy for arabinoxylan) were significantly affected by genotype (G), environment (E), and G × E interactions in high-AX, good-breadmaking-quality crosses adapted to European conditions. It has been found that the broad sense heritability for WE-pentosans (h^2^ 0.825) is significantly higher than for TO-pentosans (0.341). The amount of WE-AX and its composition were significantly affected by genotype (0.840 and 0.721), whereas the amount and composition of TO-AX were strongly affected by the environment, with h^2^ of 0.516 and 0.372, respectively [86].

Finally, Shewry et al. found statistically significant increases in the amount of arabinoxylan, as well as β-glucan, in white flour of wheats from ‘UK Heritage samples’ [82,84].

## 3. Mixed Linkage β-Glucan

The term β-glucan refers to any polymer of β D-glucose linked with glycosidic bonds. Differences in the type of glycosidic bond give these polysaccharides unique characteristics. In plants, the most abundant form of β-glucan is cellulose, consisting of linear chains of (1–4)-linked β-D-glucopyranosyl monomers, resulting in an insoluble polymer [87]. In contrast, mixed linkage β-glucan (also referred to simply as β-glucan, and referred to as such in this paper) consists of unbranched and unsubstituted blocks of β-(1–4) linked D-glucopyranosyl units (mainly cellotriosyl (G3) and cellotetraosyl (G4) blocks) linked together with β-(1,3) linkages. A β-glucan molecule can either be soluble or insoluble depending on the ratio of G3:G4 units present in the polysaccharide. For example, any long stretches consisting of only G3 or G4 could favor associative interactions with other long linear polysaccharides, causing decreased solubility [88,89]. The β-(1,3) linkages are non-randomly distributed because they never appear next to each other, but the G3 and G4 units are randomly arranged. The structural characteristics of β-glucan affects the health benefits associated with it; insoluble β-glucan affects bulking and gut motility [90,91,92], while both soluble and insoluble β-glucan can have beneficial effects on blood glucose and lipids [93,94,95,96].

β-glucan appears in members of the grasses (*Poaceae*) almost exclusively with some exceptions (i.e., *Equisetum* and some lichens) [97]. β-glucan content varies between genus, within species, between varieties, and in relation to developmental stage and growing environment [98,99,100,101]. It is suggested that β-glucans are not essential structural components of cells walls but rather represent a secondary source of metabolizable energy in the form of glucose [102,103,104]. Any attempt at adjusting the amount of β-glucan could therefore have a knock-on effect on the carbon pool in the plant. Results from different studies show variable alterations in the carbon pool especially in terms of starch content. Reduction in β-glucan content in the grains of a *Brachypodium distachyon* TILLING mutant (heterozygous for a loss-of-function mutation in *BdSclF6*) led to a more than 2.5-fold increase in starch compared to wild type [105]. These plants showed no difference in grain weight or the distribution of cellulose and xylans. Over-expression of *HvCslF6* caused an increase in β-glucan content in hull-less barley accompanied by decreased starch content in the grains, although there were also substantial changes in grain morphology [106]. In contrast, in the durum wheat Svevo-HA (a high-amylose TILLING line with knock-down alleles in two homoeologous starch branching enzyme IIa genes), β-glucan content was increased substantially (from 0.49 to 1.4%), with no significant change in total starch. However, the percentage of resistant starch was increased from 0.2 to 6.9% [107]. Therefore, the result of a targeted change in the carbon pool can be unpredictable and seems to relate to the specific point of change in the given polysaccharide synthesis pathway.

While barley and oat have the highest β-glucan content in cereal crops [43,98,108,109], wheat is far more widely consumed (Figure 1) [110] and is therefore a good target for improved β-glucan levels. Hexaploid wheat flour contains between 0.2% (endosperm flour) and 0.84% (wholemeal flour) β-glucan (Figure 4, Appendix A), which is largely insoluble [88,103,111]. Primitive wheats and wild relatives have much higher β-glucan content (up to 4.53% in *Aegilops* species) and represent a rich resource for the improvement of β-glucan content in domesticated wheat [77,78]. As stated above, the structure and solubility of β-glucan relates to its associated health benefit. Any change in content could affect G3:G4 ratio [76,106], which might affect solubility. The G3:G4 ratio in wheat differs between different tissues and milling fractions (Table 5). In general, β-glucan with very high or very low G3:G4 ratios is less soluble compared to β-glucan with a ratio of 1.5:1–2.5:1 [112,113]. This trait appears to be genetically controlled [60]; however, in vitro studies have shown that the availability of uridine diphosphate glucose (UDP-Glc) can also affect the G3:G4 ratio, where a saturating amount of UDP-Glc results in predominantly G3 units. At low UDP-Glc concentrations, the synthesis of G4 units is favored [114]. 

### 3.1. Genetic Control of β-Glucan Content and Structure

Genes in the *Cellulose Synthase Like* (*Csl*) family are responsible for the biosynthesis of non-cellulosic polysaccharides of cell walls [117]. Within this family, the *CslF*, *H,* and *J* clades have been implicated in the synthesis of β-glucan in various grass species (Figure 2B) [105,117,118,119]. Evidence has shown that the *CslF* gene family plays a dominant role in β-glucan synthesis in cereals, with *CslF6* and *CslF9* being major contributors [120,121]. The contribution of genes in the *CslH* and *CslJ* clades is less clear [118].

In hexaploid wheat, there are 10 *CslF* genes spread over the three different genomes (A, B, and D) [61]. *CslF6* is the only member of the *CslF* subfamily that is highly expressed in grain [61]. The key role of *CslF6* has been illustrated through studies which showed that decreased gene expression leads to decreased β-glucan content in wheat [121] and that mutation or knock-out in this gene leads to essentially β-glucan-less grain in barley [119,122]. While genetic differences have been identified within the *CslF6* gene and promoter regions in barley, these polymorphisms could not be correlated with differences in β-glucan content [119,123,124]. The relationship between transcript abundance and β-glucan content has also been studied well in barley [124,125]. Studies confirm that *CslF9* expression peaks at around 8–10 days post-anthesis [124,125], with *CslF6* differentially expressed at the same time but also later in grain development at 38 dpa [124]. In contrast to the crucial role of *CslF6* in β-glucan content in barley, *CslF9* knock-out mutants did not show a difference in β-glucan content compared to controls, although there was a significant decrease in starch compared to wild type [119], which underscores the connection between β-glucan and starch content in the wheat grain. The involvement of *CslH* in the synthesis of β-glucan has been demonstrated through heterologous expression in Arabidopsis [118], while heterologous expression of *CslJ* in *N. benthamiana* also produced β-glucan, even in the absence of other members of the *Csl* gene family [126]. The *CslH* and *CslJ* gene families are much smaller than the *CslF* gene family, with a genome-wide analysis suggesting only eight proteins in the *TaCslH* subfamily and four proteins in the *TaCslJ* family (compared to twenty-nine proteins in the *TaCslF* subfamily) [61]. Investigation of publicly available RNA-Seq datasets revealed low expression of *CslJ* and *CslH* in wheat grain [61]. A quarter of the 108 *Csl* genes identified in wheat are predicted to have two or three splice variants, with alternative splicing leading to ten splice variants in *TaCslF*, three in *CslH,* and four in *CslJ* [61]. Transcript abundance and β-glucan synthase activity does not necessarily correlate with β-glucan content in barley varieties. In addition, β-glucan content can decrease from previous levels upon grain maturation, supporting evidence that other factors, such as the availability of UDP-Glc and β-glucan endohydrolase activity, could play a role [124,127].

Studies investigating changes in β-glucan structure have identified the role of *CslF6* in regulating G3:G4 ratio, and hence physicochemical characteristics of the polysaccharide. Species-specific residues in the CslF6 protein have been linked to characteristic structural differences in the resultant β-glucan [60,128]. A single amino acid change in CslF6 (I571L) caused an increase in the proportion of β1–4 bonds in the β-glucan synthesized. The mutation relates to a trans-membrane helix, suggesting that the change affects the movement of the growing β-glucan chain within the membrane channel [60]. Amino acid substitutions in the catalytic region of CslF6 changed both the structure (CslF6 G638D in *Sorghum bicolor*) and amount (CslF6 Y680F in *S. bicolor*) of β-glucan when transiently expressed in *N. benthamiana* [128]. These amino acid changes are close to conserved regions of the enzyme and are postulated to affect conformation of conserved regions in the catalytic region, affecting the orientation of the nascent polysaccharide acceptor and the UDP-Glc donor.

Co-expression of genes other than those directly linked to β-glucan synthesis have been observed and provide evidence for the involvement of *trans* elements in regulating β-glucan content in cultivars that show variation in this trait [120,125]. These include *HvGlb1* in barley, which encodes a β-glucanase isoenzyme I, which also correlates with malt β-glucan content and malt quality parameters [129]. While a lot of evidence points to the role of specific genes in determining β-glucan content, much remains to be explored. An increasingly clear understanding of the genetic control of β-glucan content and structure will support a targeted approach in manipulating this trait in wheat cultivars.

Genes involved in β-glucan biosynthesis are summarized in Table 2.

### 3.2. QTLs Linked to β-Glucan Content

Quantitative trait locus (QTL) analyses and genome-wide association studies have been performed on wheat to find markers that could be used for selection of varieties with improved β-glucan characteristics. In tetraploid wheat, QTLs associated with grain β-glucan content have been identified on chromosomes 1A, 2A, 7A, 2B, and 5B (Table 6). Genes associated with these markers, and that showed detectable expression in grain caryopsis, embryo, and endosperm, were identified as starch synthase II, β-amylase, isoamylase, fructan 1-exohydrolase, and (1,4)-β-xylanase [130]. This study, interestingly, did not identify any genes in the *CslF* or *CslH* family, which are known to be involved in β-glucan synthesis. However, the associated genes play roles in carbon partitioning (particularly starch synthesis), further supporting a link between the biosynthetic pathways of these two cell wall components [105,131]. Furthermore, it is assumed that β-glucan synthesis involves the interaction of *CslF* and *CslH* with other proteins, such as those identified in this study or elements such as transcription factors [132,133]. In a separate study by the same group, new QTLs for β-glucan were identified on chromosomes 2A and 2B, also in tetraploid wheat [62]. 

In hexaploid wheat, QTLs have been identified on chromosomes 3A, 1B, 5B, and 6D (Table 6) [134]. The QTL on chromosome 5B showed large phenotypic variation and was attributed to the parental plant *T. aestivum* cv Chinese Spring (the other parent plant was *T. spelta* var duhamelianum KT19-1). However, this marker could not be successfully mapped, while the QTL on chromosome 3A was related to a glucan endo-1,3-β-glucosidase. Recently, three QTLs related to β-glucan content were identified in *Ae. Biuncialis* in an effort to discover more accessions that are suitable for interspecific hybridization programs [78]. These were found on chromosomes 1M, 4M, and 5M. While *Csl* genes were not identified as candidates, markers were identified on the same homoeologous group chromosomes that have *Csl* genes assigned to them in wheat (groups 1 and 5) (Table 6). Most of the wheat QTLs associated with β-glucan content are not stable across environments [62,134]. Marker-assisted selection has so far not been used to improve β-glucan content in wheat due to a paucity of major, stable QTLs identified for the trait. This is in contrast with other cereals, e.g., barley [63,135,136,137,138] and oat [139,140].

### 3.3. Breeding Approaches to Improve β-Glucan

Traditional breeding methods require access to populations with very broad genetic variability [141]. In the absence of this variability, wide crosses (hybridization between different species) and other techniques can be considered. Alien introgression—the addition of a single chromosome pair from a donor to an otherwise incompatible acceptor plant—allows incorporation of a trait into hexaploid wheat (AABBDD genome). Occasionally, the additional chromosome pair from the donor will substitute for the native homoeolog and lead to a substitution line. Alternatively, amphidiploid hybrids can be backcrossed to the wheat parent to produce wheat lines that contain a single chromosome or chromosome pair from the donor parent [142,143,144,145,146]. The addition of a single chromosome from another species allows studying of the genetic effect of the individual chromosome in the wheat genomic background. However, the trait is often unstable and lost through segregation or affected by the inclusion of gametocidal genes in the transferred chromosome [147,148]. General breeding approaches to improve β-glucan are summarized in Figure 3.

#### 3.3.1. *Triticum* and *Aegilops* Species

In a large-scale screen of β-glucan content in 500 wheat accessions (including hexaploid, tetraploid, wheat–barley addition lines, and triticale lines) it was concluded that there is insufficient genetic diversity in wheat germplasm to initiate a breeding program aimed at obtaining a target β-glucan content of 20 g kg^−1^ in wheat grain (the minimum level considered sufficient for a reduction in blood cholesterol) [149]. Even though small-seeded primitive grains showed the greatest variation in β-glucan content in this study, the higher β-glucan content trait was not transferred to synthetic hexaploid wheats with a primitive line as parent. In synthetic wheat lines, the trait from the tetraploid parent (low β-glucan, 0.49 %dw on average) was observed, although there were several diploid parental lines with high β-glucan content such as *Aegilops squarrosa* (1.8% dw, DD genome) and *Aegilops speltoides* (1.68% dw, BB genome) [149]. In contrast, a smaller panel-wide analysis of β-glucan content in wheat accessions identified two *Aegilops* species (*Ae. umbellulata*, UU genome and *Ae. markgrafii*, CC genome) as potential candidates for improvement of β-glucan in wheat [77]. This study analyzed a much wider range of *Tritiaceae* genomes including species containing the D, S, U, T, and M genomes. On average, *Aegilops* species showed a higher β-glucan content than *Triticum* species in this study. In particular, species with the U genome appeared to have a substantially higher β-glucan content (5.3 ± 1.4% *w*/*w* on average between six species over 2 years) compared to cultivated hexaploid bread wheat (0.83 ± 0.09% *w*/*w* on average between three varieties over 2 years). A significant negative correlation was also observed between kernel weight and β-glucan content when considering all the species assessed in this study, although this correlation changed to positive when considering only *Aegilops* species. This is most likely due to the size of *Aegilops* kernels compared to *Triticum* kernels, which have a much larger proportion of endosperm.

In agreement with Marcotuli et al. (2019) [77], Rakszegi and co-workers (2017) [76] illustrated the utility of *Aegilops* species in the improvement of β-glucan content in wheat (Table 4). They found that the addition of specific chromosomes (namely 5U^g^, 7U^g^ of *Ae. geniculata* and 7M^b^ of *Ae. biuncialis*) to bread wheat led to an increase in β-glucan under optimal and drought conditions compared to control cv. Chinese Spring. In these lines, the thousand kernel weight (TKW) was unchanged or higher compared to the control. The results were confirmed under field conditions. There was also an impact on the ratio of G3:G4 in some of the addition lines, where the addition of 5U^g^, 6U^g^, or 3U^b^ decreased the ratio of G3:G4. *Aegilops* homologues of *CslF* and *CslH* were assigned to the same homeologous group chromosomes (group 1, 2, 5, and 7) as in bread wheat based on comparative analyses using cDNA sequences from other grass species and chromosome survey sequences of the *Ae. umbellulata* genome. While *Ae. umbellulata* has a homologue for *CslF6* on chromosome 7UL, chromosome 5U carries a copy of *CslF7* and 6U carries *CslF11*. These addition lines contained whole homeologous chromosome pairs from *Aegilops* and may therefore include genes that negatively affect wheat grain characteristics. In a separate study where β-glucan content was not analyzed, an addition line was generated between wheat (*T. aestivum* var. Chinese Spring, AABBDD) and *Ae. umbellulata* (UU) carrying a pair of 1U chromosomes. The addition line showed increased total protein in grain, improved dough quality, and overall improved agronomic traits, providing evidence for the potential of this species as a source of genetic material for the improvement of bread wheat [150]. 

#### 3.3.2. Other Members of the *Triticeae* Species

Barley (HH genome), oats (AADDCC genome), and rye (RR genome) contain the highest amount of β-glucan compared to other cereals [43,98,108] and represent valuable genetic resources to increase β-glucan in wheat. Hexaploid tritordeum (AABBHH) is an amphidiploid hybrid created by crossing durum wheat (*Triticum durum*) with wild barley (*Hordeum chilense*) followed by colchicine treatment to double chromosome number to 42 and improve fertility [151]. Since the development of this crop species, its beneficial properties and potential as a bridge to introduce traits from *H. chilense* to wheat have been widely investigated [152,153,154,155]. While Tritordeum has five times less β-glucan compared to barley, it has twice the amount present in durum wheat and the same amount as soft wheat [153]. It is encouraging to see that this hybrid has double the β-glucan content compared to the lower β-glucan-containing parent.

Asakaze/Manas wheat–barley hybrid panels exist consisting of disomic and ditelosomic addition lines where the wheat line (Asakaze) contains either a whole barley chromosome, or the short or long arms of chromosomes 2, 3, 4, 6, or 7, respectively [156,157]. From this panel of addition lines, a line showing elevated β-glucan content (Asakaze/Manas 7H, 2n = 44) was used as a breeding partner with a winter wheat 7B monosomic line (cv Rannaya, as female parent) to create a compensating wheat/barley Robertsonian translocation line (7BS.7HL centric fusion, 2n = 42). The 7H disomic addition line was chosen since barley chromosome 7H carries the *HvCslF6* gene, which is directly tied to the synthesis of β-glucan. While the 7H addition line showed low fertility, the Robertsonian translocation line showed similar fertility to the Rannaya winter wheat cultivar. However, the 7BS.7HL translocation line showed a decreased β-glucan content (0.9%) compared to the 7H addition line (1.1%), although both had a larger β-glucan content compared to the wheat cultivar Rannaya (0.7%). The β-glucan content in the barley parent (approximately 4.9%) was significantly different from all other lines and this trait was not conferred by a simple transfer of the *HvCslF6* gene to a wheat line, supporting the hypothesis that β-glucan content is determined by several factors.

The wheat–barley addition lines above reflected the β-glucan content observed in the parental line with the lower content as opposed to the high-β-glucan donor. However, introgression of the barley 7H chromosome (particularly the long arm, which contains the *HvCslF6* gene) into recombinant chromosomes of the A, B, and D genome of wheat individually led to a 0.8–1% increase in measured β-glucan in wheat compared to control [158]. A further experiment [159] showed that increasing the number of barley *HVCsfl6* copies in wheat (through more targeted introgression of 7HL) could significantly increase the β-glucan content, supporting a direct link between β-glucan content and gene copy number. This result indicated the importance of the *CslF6* gene on the D genome and confirmed results that show higher β-glucan content in hexaploid wheat (AABBDD) compared to tetraploid wheat (AABB), and higher β-glucan content in *Ae. tauschii* (DD) compared to *T. urartu* (AA) and *T. monococcum* (AA) [77]. In another study on the effect of the 7H chromosome in wheat–barley addition lines, variable results were observed, with individual addition lines showing increased and decreased β-glucan content [160]. In some instances, the environmental conditions would affect the amount of β-glucan measured over different years. However, one addition line (CS-7H) showed a consistent increase in β-glucan content compared to wheat control over all years. These results supported previous studies showing increased β-glucan in wheat containing the barley 7H chromosome [158,159,161]. In addition, a ditelosomic wheat addition line containing the 1HS ditelosome (containing the *HvCslF9* gene) showed a significantly increased β-glucan content [161], confirming the importance of this gene. While β-glucan increased in some of these addition lines, the overall β-glucan content was still lower than that observed in barley, implying that expression level of β-glucan is a much more complicated pathway, either requiring other genes, or being regulated or out-competed for substrate.

Although rye has a high β-glucan content, few studies have investigated this species as a breeding partner for improving β-glucan content in wheat. Triticale (AABBRR) is a hybrid of wheat (*Triticum turgidum*) and rye (*Secale cereale*), produced to combine the high yield potential and grain quality of wheat with the abiotic and biotic stress resistance of rye [162]. While whole rye flour has a β-glucan content of 1.0–2.5% [163], the content was measured in triticale at 0.35–0.96% [149], suggesting that the trait was not transferred, although the higher values represent a significant increase over the β-glucan content in *T. turgidum* (0.41%) [130].

### 3.4. Heritability of the β-Glucan Trait

The broad sense heritability of the β-glucan content trait has been observed at 0.80–0.82 in tetraploid wheat, indicating that the phenotype potentially has a strong genotypic basis [62,130]. In *Ae. biuncialis,* the heritability was shown to be even higher, at 0.93 [78]. However, the heritability in a different GxE study showed heritability at 0.51 [43] and a significant interaction between genotype and year of growth was observed in a panel of *Triticum* and *Aegilops* genotypes [77]. Other factors, such as variety and growth conditions, can also affect β-glucan content. For example, in one study, barley β-glucan content varied from 3.91 to 5.93% (dry weight) between varieties [164]. In another comprehensive study assessing 17 varieties of barley over eight sites and 2 years, barley β-glucan content ranged from 1.81% to 7.18% (*w*/*w*) between varieties. Even in a single variety, β-glucan varied by as much as 2.61% (where the average β-glucan content for all varieties was 3–4%) [165]. The authors concluded that rainfall, location, and genotype influence β-glucan content, with significant interactions between genotype and year and genotype and location on this trait [165]. Environmental conditions also affect the ratio of β-glucan and AX, with heat and drought generally decreasing β-glucan with a concomitant increase in AX (although this is variety-dependent) [101].Genetic approaches to improve β-glucan content in wheat lines therefore have to take into consideration environmental interactions on the given genotype.

## 4. Conclusions

Historically, improving traits in wheat has been challenging due to the large genome size (16 Gb), its polyploid nature, and large portion of repetitive sequences. Since 2014, when the first genome assembly for Chinese Spring wheat was produced by the International Wheat Genome Sequencing Consortium (IWGSC) [166], several others have been released and all set down the basis for understanding the genetic diversity of this important crop. With the ever increasing number of resources and approaches used in wheat research, we have now generated a vast amount of data that need to be utilized in wheat breeding programs to not only increase the yield and resistance to ever changing environmental stresses, but also to improve the nutritional value that can be tailored to the masses to lower the burden on our healthcare systems and simultaneously increase the quality of life.

Arabinoxylan has been in the spotlight of several academic research groups. Identification of GTs, QTL mapping, and establishing the markers to select varieties with improved AX properties has been the priority reaching far beyond the last decade. Also, for developing and characterizing crosses with increased amounts of AX in wheat grains, the work on exploiting the genetic variation controlling the content of AX in wheat grain is hardly finished. 

Although several attempts have been made to increase the content of β-glucan in wheat, the complexity of the trait and the lack of a major, stable QTL has impeded progress. In several breeding studies, the β-glucan content of offspring seems to reflect that of the parent with the lower β-glucan value, showing that the high-content trait is not always expressed in the offspring. While the addition of barley chromosome 7 or chromosomes from the U genome of *Aegilops* species has shown a beneficial effect, environmental interactions mean that the trait is not necessarily stably expressed and can still vary substantially year on year in these lines.

Nevertheless, strategies and challenges to manipulate the amount of dietary fiber and how to translate these to improve wheat for human consumption and health without negative effects on cost, consumer palatability, and processing properties need to be discussed and implemented simultaneously.

Whereas the amount of AX in white flour seems to follow the increasing trend in 39 UK-adapted winter bread wheat cultivars dating between 1790 and 2012, the amount of β-glucan varies more and shows a weaker trend, except that it is low in the old cultivars. Nevertheless, the absolute quantity of both components was higher in modern cultivars (post 1982) but still heavily affected by the environment [82].

Finally, beyond improving dietary fiber content in wheat, it is also important to determine if increases in dietary fiber content translate to measurable health benefits in vivo. The development of standardized in vitro protocols [167,168] and in silico models like the Dynamic Gastric Model [169] have made it possible to predict physiological relevance in humans and pave the way for in vivo studies. As one example for all, Gouseti et al. have demonstrated that exploiting genetic variation of the dietary fiber amount in wheat cultivars is a possibility, resulting in the production of high-fiber white breads that are healthy, with reduced starch digestion rate yet acceptable to consumers [170].

## Figures and Tables

**Figure 1 plants-12-03216-f001:**
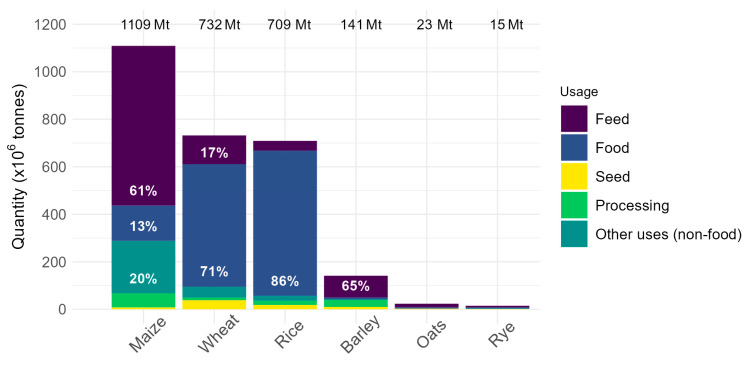
Usage of the top six cereal crops worldwide (FAO.org, accessed on 12 July 2023). The use of cereals as feed for animals (purple), food for humans (blue), and seed for propagation (yellow), as well as in processing (green) and non-food applications such as industry and biofuels (blue-green).

**Figure 2 plants-12-03216-f002:**
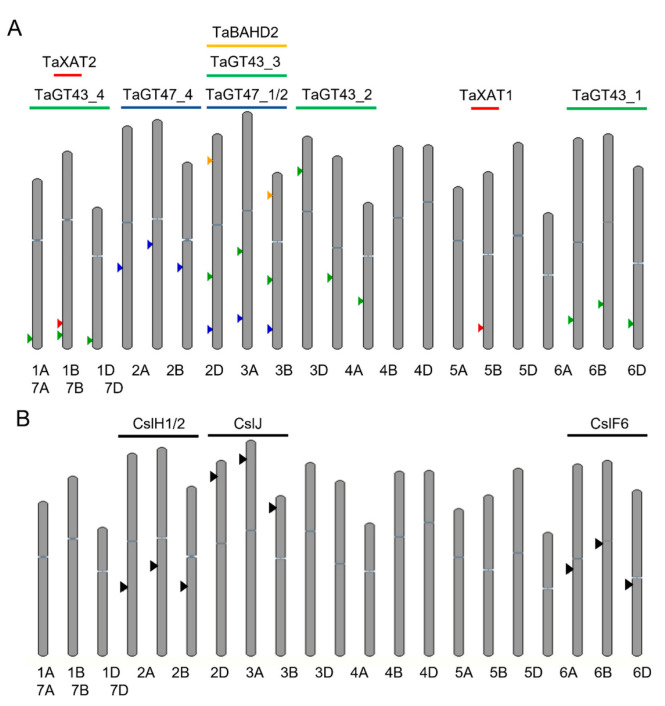
Position of genes involved in the biosynthesis of arabinoxylan (**A**) and β-glucan (**B**) in the A, B, and D genomes of hexaploid wheat. Arrows indicate position of gene as labelled above. Gene details are as in Table 2.

**Figure 3 plants-12-03216-f003:**
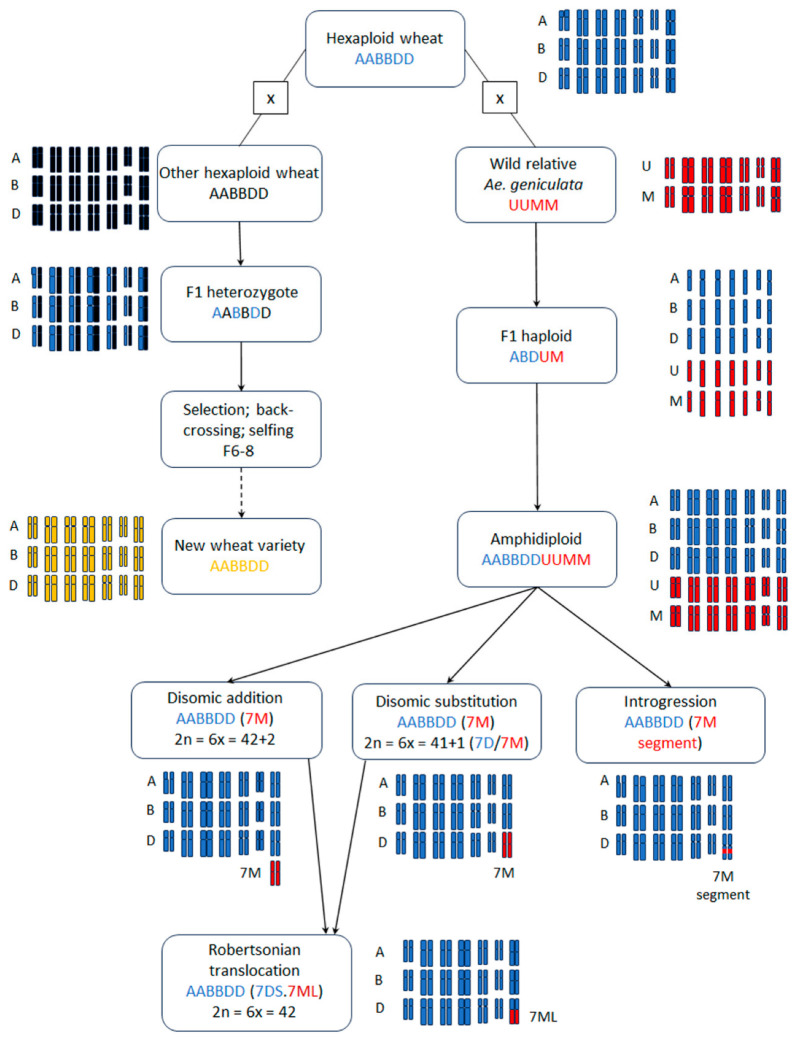
Breeding approaches employed to increase the amount of dietary fiber in wheat. Hexaploid wheat can be crossed with hexaploid wheat in a traditional breeding approach, or other germplasm (such as wild relatives or other domesticated cereals) can be utilized. Here, the example of *Ae. geniculata* is used.

**Figure 4 plants-12-03216-f004:**
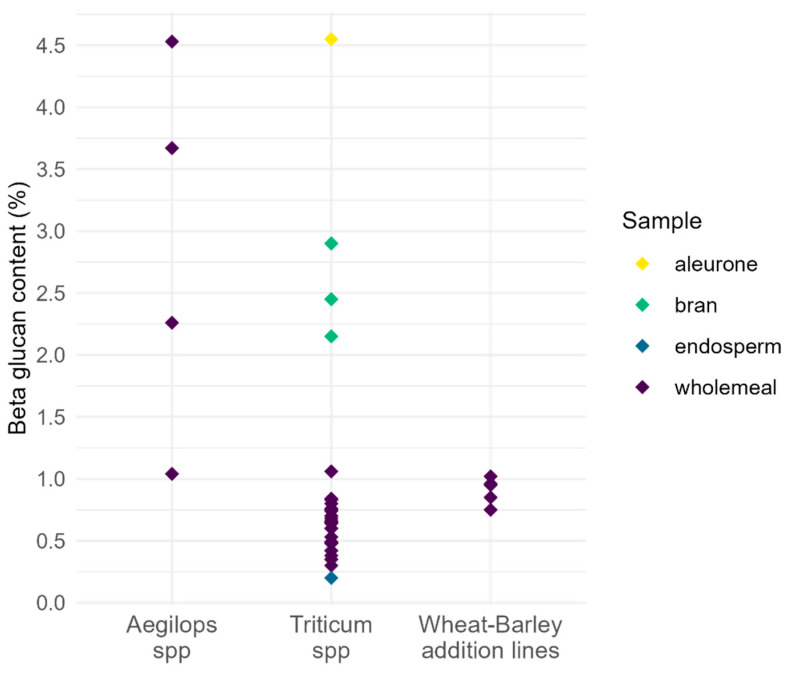
β-glucan content as measured in *Aegilops* and *Triticum* species, as well as wheat–barley addition lines from various studies (see Appendix A for specific data). Colors represent flours made from different parts of the cereal grain.

**Table 1 plants-12-03216-t001:** The AX content in wholemeal and flour of wheat and other major cereals.

	TO-AX [%]	WE-AX [%]	n	Analytical Method	Reference
**Wholemeal**					
wheat (%dw)	6.36	0.56	22	GC	[37]
wheat (%dw)	5.80	-	26	GC	[38]
wheat (%dw)	6.20	-	1	Uppsala method	[39]
winter wheat	1.90	0.50	131	GC	[40]
spring wheat	2.00	0.50	20	GC	[40]
durum wheat (%dw)	4.06	0.57	15	HPAEC	[41]
durum wheat	1.95	0.40	10	GC	[40]
spelt	1.75	0.35	5	GC	[40]
einkorn	1.95	0.60	5	GC	[40]
emmer	1.70	0.25	5	GC	[40]
rye (%dw)	6.69	3.89	2	GC	[42]
rye (%dw)	7.93	-	5	GC	[43]
rye (%dw)	8.60	-	1	Uppsala method	[39]
triticale (wheat × rye, %dw)	6.70	-	8	Uppsala method	[39]
spring barley (dw)	8.10	-	20	Uppsala method	[39]
tritordeum (durum wheat × wild barley, %dw)	6.90	-	5	Uppsala method	[39]
oat (%dw)	11.60	0.90	1	HPAEC	[44]
oat (%dw)	8.35	-	141	HPAEC	[45]
**Flour**					
wheat (%dw)	2.18	0.51	20	GC	[46]
wheat (%dw)	1.99	0.54	26	GC	[38]
wheat (synthetic × Opata cross, %dw)	2.35	0.51	90	GC	[47]
rye (%dw)	3.64	1.21	11	GC	[48]
rye (%dw)	3.12	1.36	5	GC	[43]
spring barley (%dw)	1.93	0.23	6	GC	[49]
winter barley (%dw)	1.88	0.27	4	GC	[49]
oat (%dw)	3.30	-	1	HPAEC	[45]

**Table 2 plants-12-03216-t002:** Summary of genes involved in biosynthesis of arabinoxylan and β-glucan (* notes genes with the highest transcript abundance in developing wheat grain within the class, as in [50]).

Polysaccharide	Protein Function	Gene Name and ID (if Known)	Reference
AX	backbone synthesis (GT43)	*TaGT43_1 ** TraesCS7A02G441400, TraesCS7B02G340100, TraesCS7D02G430700	[50,53,59]
		*TaGT43_2 ** TraesCS4A02G107400, TraesCS4B02G197000, TraesCS4D02G197300	
		*TaGT43_3* TraesCS3A02G270100, TraesCS3B02G304000, TraesCS3D02G269800	
		*TaGT43_4* TraesCS1A02G391000, TraesCS1B02G419100, TraesCS1D02G399000	
	backbone synthesis (GT47)	*TaGT47_1* TraesCS3A02G440100, TraesCS3B02G474200, TraesCS3D02G432900	
		*TaGT47_2 ** TraesCS3A02G440800, TraesCS3B02G474900, TraesCS3D02G433400	
		*TaGT47_4* TraesCS2A02G288300, TraesCS2B02G305100, TraesCS2D02G286600	
		*TaGT47_3*, *TaGT47_5*, *TaGT47_6*, *TaGT47_10*, *TaGT47_13*, *TaGT47_14*	
	arabinosylation (GT61)	*TaGT61_1/TaXAT1 ** TraesCS6A02G309400	[50,57]
		*TaGT61_2/TaXAT2* TraesCS1B02G371300	
		*TaGT61_9*, *TaGT61_11*, *TaGT61_13*, *TaGT61_14*	
	feruloylation (BAHD)	*TaBAHD1 **, *TaBAHD3 **, *TaBAHD4*, *TaBAHD5*	[50,54]
		*TaBAHD2* TraesCS3A02G119500, TraesCS3A02G119700, TraesCS3D02G121800	
β-glucan	synthesis/regulating G3:G4 ratio	*TaCslF6 ** TraesCS7A02G298600, TraesCS7B02G188400, TraesCS7D02G294300	[60,61,62,63]
	synthesis	*CslH1* TraesCS2A02G302300, TraesCS2B02G318100, TraesCS2D02G300900	[61,62,63]
	synthesis	*TaCslJ1/2* TraesCS3A02G094600, TraesCS3B02G110100, TraesCS3D02G095600, TraesCS3D02G094800	[61]

**Table 3 plants-12-03216-t003:** QTLs identified in wheat related to arabinoxylan content, trait (Wf = white flour; Wm = wholemeal; RV = relative viscosity; WE-AX = water-extractable AX; TO-AX = total AX; WU-AX = water-unextractable AX), chromosome position (Chr), LOD score, and candidate genes. (* major QTL identified in the study).

Plant Material	Trait	QTL ID	Chr	Nearest Marker or Interval	LOD Score	Candidate Genes	Ref
*T.aestivum* (bread wheat)	Wf RV/WE-AX	1-ARE	1B		>4.7		[65]
		2-CtCs	1B		>3.8		
	Wf WE-AX	3-R6C7	1B		14.5		
	Wf RV/WE-AX	4-RER	3B		3.74		
		5-CtCs	3D		3.99		
		6-RER	3D		4.96		
		7-CtCs	4B		3.81		
		8-CtCs	5D		26.8		
		9-CtCs	6B		3.82		
	Wf WE-AX	10-R6C7	6B		16.4		
	Wf RV	11-VxI*	6B		16.5		
	Wf RV/WE-AX	12-RER	7A		13.9		
	Wf RV	MQTL1	1B	Xwpt5061	7.63	ribosomal protein	
		MQTL2	3D	XksuD14	17.7	kinase inhibitor	
		MQTL3	6B	Xwpt-8641	2.61	translation initiation factor	
*T.aestivum* (bread wheat)	Wm TO-AX	QGax.aww-2A.1 *	2A	wpt-3114-2A			[66]
		QGax.aww-3D.1	3D	wpt-0485-3D			
		QGax.aww-4D.1 *	4D	gpw-95001-4D			
		QGax.aww-6B.1	6B	gwm680-6B			
*T.durum* (tetraploid wheat)	Wm TO-AX	QGax.mgb-1A.1	1A	wsnp_Ex_c45880_51550172		GH47, Gal7/GH35	[67]
		QGax.mgb-1A.2	1A	RFL_Contig399_976		GT31	
		QGax.mgb-1B.1	1B	Ex_c40520_1484			
		QGax.mgb-1B.2	1B	BS00039135_51			
		QGax.mgb-2A.1	2A	BS00073381_51			
		QGax.mgb-2A.2	2A	GENE-0762_808			
		QGax.mgb-2B.1	2B	Tdurum_contig45838_263		TaUGT1/GT1, cisZog2B/GT1, GT4	
		QGax.mgb-3A.1	3A	Kukri_c17966_634		CelC/GH1	
		QGax.mgb-3B.1	3B	GENE-4918_283			
		Qgax.mgb-4B.1	4B	Tdurum_contig42229_113			
		QGax.mgb-5A.1	5A	Ex_c95453_1499		GT8, Ugt12887/GT1	
		QGax.mgb-5A.2	5A	BS00068254_51		GT2, CE8	
		QGax.mgb-5A.3	5A	tplb0056b09_1000		TaUGT1/GT1	
		QGax.mgb-6A.1	6A	BobWhite_c27145_318			
		QGax.mgb-6B.1	6B	BS00063217_51			
		QGax.mgb-7A.1	7A	Tdurum_contig69003_459		Gsl12/GT2/GT48 (β-1,3-glucan synthase)	
		QGax.mgb-7A.2	7A	wsnp_Ex_c21854_31021668		Cel8/GH9	
		QGax.mgb-7A.3	7A	GENE-4672_55			
		QGax.mgb-7B.1	7B	Kukri_c42653_179			
*T.aestivum* (bread wheat)	Wm TO-AX	QgTOT-AX.caas-1B	1B	HVM23–Sec1	10.5		[68]
		QgTOT-AX.caas-1D	1D	Xwmc336–Xbarc152	3.1		
		QgTOT-AX.caas-3B	3B	Xbarc115–Xbarc344	2.9		
		QgTOT-AX.caas-5B	5B	Xbarc142–Xwmc28	3.3		
	Wm WU-AX	QgWU-AX.caas-1B	1B	HVM23–Sec1	5.5		
		QgWU-AX.caas-3B	3B	Xbarc115–Xbarc344	4.2		
	Wm WE-AX	QgWE-AX.caas-1A	1A	Xbarc148–Xwmc449	6.8		
		QgWE-AX.caas-1B	1B	HVM23–Sec1	10.5		
		QgWE-AX.caas-2B	2B	Xwmc441–Xcfe52	9.2		
		QgWE-AX.caas-3B	3B	Xbarc115–Xbarc344	3.9		
		QgWE-AX.caas-5A	5A	Xgwm443–Xcwem44	4.1		
		QgWE-AX.caas-5B	5B	Xbarc142–Xwmc28	8.7		
		QgWE-AX.caas-6B	6B	Xbarc79–Xbarc178	3.7		
		QgWE-AX.caas-7A	7A	Xbarc174–Xbarc108	3.3		
		QgWE-AX.caas-7B	7B	Xbarc1181–Xwmc517	6.5		
*T.aestivum* (bread wheat)	Wf TO-AX	Y34Val-1A	1A	AX-94522489	2.4		[69]
		Y34Ukr-1A *	1A	AX-94902531	3.2		
		Y34Cla-1B *	1B	AX-94385888	3.2		
		Y34Val-1B	1B	AX-94524314	2.5		
		Y34Ukr-1B *	1B	AX-94845742	5.1		
		Y34Ukr-2A	2A	AX-95164135	2.9		
		Y34Cla-2D	2D	AX-94538798	2.5		
		Y34Cla-5D	5D	AX-94877826	1.6		
	Wf RV/WE-AX	Y34Val-1A *	1A	AX-94430904	3.8		
		Y34Alt-1B *	1B	AX-94618000	12.6		
		Y34Val-1B *	1B	AX-94807857	7.8		
		Y34Cla-2B *	2B	AX-94421649	3.1		
		Y34Alt-2B	2B	AX-94546045	2.7		
		Y34Alt-2D *	2D	AX-94452103	3.2		
		Y34Cla-3A	3A	AX-94603083	2		
		Y34Alt-3B *	3B	AX-94382595	6.1		
		Y34Ukr-3B	3B	AX-94769959	2.9		
		Y34Cla-3B *	3B	AX-95629178	5.9		
		Y34Alt-4B	4B	AX-94853726	2.4		
		Y34Alt-4D *	4D	AX-94766682	3.5		
		Y34Val-6B *	6B	AX-94593804	4.4		
*T.aestivum* (bread wheat)	Wf TO-AX	1	1B	1B_646895451		TraesCS1B02G424500/GH16	[70]
		2	1B	1B_653086336			
		3	1B	1B_653681771		TraesCS1B02G429500/GT61	
		4	1B	1B_654915479			
		5	5B	5B_14665450			
	Wf WE-AX	6	1B	1B_646895451		TraesCS1B02G424500/GH16	
		7	1B	1B_653086336			
		8	1B	1B_653681771		TraesCS1B02G429500/GT61	
		9	1B	1B_654915479			
		10	2B	2B_184634480		TraesCS2B02G204300/GH43	
		11	6B	6B_26597224			
		12	7A	7A_234827309		TraesCS7A02G250500/peroxidase TraesCS7A02G251400/GH13/peroxidase	
		13	7A	7A_264333614			
		14	7A	7A_458678969		TraesCS7A02G317700/GH9 TraesCS7A02G319100/peroxidase	
		15	7A	7A_474572231			
		16	7A	7A_516508921		TraesCS7A02G349200/GH11 TraesCS7A02G352000/peroxidase TraesCS7A02G352900/peroxidase TraesCS7A02G353000/peroxidase TraesCS7A02G353200/peroxidase TraesCS7A02G353300/peroxidase TraesCS7A02G353400/peroxidase	
		17	7A	7A_700824770		TraesCS7A02G514300/GT1	
		18	7B	7B_454100716			

**Table 4 plants-12-03216-t004:** Examples of wheat germplasms with great potential in genetic improvement of arabinoxylan (AX) and β-glucan.

Germplasm or Cross	Change in AX/β-Glucan Amount	Reference
**AX**		
Yumai34 × Ukrainka	~+ 5–9 mg/g TO-AX compared to cv Ukrainka~ + 3–4 mg/g WE-AX compared to cv Ukrainka	[71]
Yumai34 × Lupus	~+ 3–4 mg/g TO-AX compared to cv Lupus~+ 2–3 mg/g WE-AX compared to cv Lupus
*Aegilops geniculata*Addition line:5U7U	less TO-AX, more WE-AX compared to cv Chinese Spring	[76]
+7 mg/g compared to control+7 mg/g compared to control
*Aegilops biuncialis*Addition line:1U	less TO-AX, more WE-AX compared to cv Chinese Spring
+5 mg/g compared to control
**β-glucan**		
*Aegilops umbellulata* (2n = 2x = 14, UU)	+62 mg/g compared to cv Chinese Spring (1 year)	[77]
*Aegilops markgrafii*(n = 2x = 14, CC)	+37.4–36.7 mg/g compared to cv Chinese Spring (2 years)
*Aegilops biuncialis*(2n = 4x = 28, U^b^U^b^M^b^M^b^)	+26.68–28.66 mg/g compared to control wheat (2 years)	[78]
*Aegilops geniculata*(2n = 4x = 28, U^g^ U^g^M^g^M^g^)Addition line:5U7U7M	~+43 mg/g compared to control wheat	[76]

+4 mg/g compared to control+4 mg/g compared to control+2 mg/g compared to control
*Aegilops biuncialis*(2n = 4x = 28, U^b^U^b^M^b^M^b^)Addition line:7M	~+20 mg/g compared to control wheat

+4 mg/g compared to control

**Table 5 plants-12-03216-t005:** G3:G4 ratio in wheat samples from selected publications (2000–2017). All analyses were performed using HPAEC-PAD.

Sample	G3:G4	Other Info	Source
Immature endosperm (17 dpa)	1.2	cv. Cadenza	[50]
Immature endosperm (21 dpa)	1.2	cv. Cadenza	[50]
Immature endosperm (42 dpa)	1.3	cv. Cadenza	[50]
Immature endosperm (28 dpa)	1.3	cv. Cadenza	[50]
Immature endosperm (35 dpa)	1.4	cv. Cadenza	[50]
Immature endosperm (14 dpa)	1.4	cv. Cadenza	[50]
Wholemeal flour	1.4	Chinese Spring 5U^g^ addition line; estimated from graph	[76]
Immature endosperm (21 dpa)	1.5	cv. Hereward	[28]
Wholemeal flour	1.6	*Ae. biuncialis*; estimated from graph	[76]
Break 1 milling fraction	1.8	cv. Hereward	[28]
Wholemeal flour	1.9	*Ae. geniculata*; estimated from graph	[76]
Wholemeal flour	1.9	Chinese Spring 6U^g^ addition line; estimated from graph	[76]
Reduction 1 milling fraction	1.9	cv. Hereward	[28]
Wholemeal flour	2.0	Chinese Spring 3U^b^ addition line; estimated from graph	[76]
Wholemeal flour	2.2	cv. Chinese Spring; estimated from graph	[76]
Wholemeal flour	2.3	Bread wheat (high nitrogen)	[115]
Wholemeal flour	2.4	Bread wheat (low nitrogen)	[115]
Wholemeal flour	2.5	cv. Hereward	[28]
Fine bran milling fraction	2.6	cv. Hereward	[28]
Coarse bran milling fraction	3.1	cv. Hereward	[28]
White wheat bran powder 50	4.3	Purified beta glucan; mean of 7 fractions	[111]
Wheat bran	4.5		[116]

**Table 6 plants-12-03216-t006:** QTLs identified in wheat relating to β-glucan content. Chromosome position (Chrs), LOD score, and candidate genes are listed.

Plant Material	QTL	Chrs	Closest Marker	LOD Score	Candidate Gene	Ref
*T.aestivum* × *T. spelta* RIL (F_8_)	QBgn	3A	Xbarc45	2.83	glucan endo-1,3-β-glucosidase	[134]
	QBgn	1B	Xhbg406	3.31	-	
	QBgn	5B	Xgwm540	5.31	-	
	QBgn	6D	Xcfd80	3.07	-	
*T. turgidum* L. ssp: *durum*, *turanicum*, *polonicum*, *turgidum*, *carthlicum*, *dicoccum*, *dicoccoides*, *aethiopicum*	QGbg.mgb-1A.1	1A	IWB42976	3.2	-	[130]
	QGbg.mgb-1A.2	1A	IWB45341	2.8	endo-β-1,4-glucanase	
	QGbg.mgb-2A.1	2A	IWB66738	3.3	starch synthase II	
	QGbg.mgb-2A.2	2A	IWB26593	3.1	b-amylase	
	QGbg.mgb-2B	2B	IWB1898	3.5	(1,4)-b xylanase	
	QGbg.mgb-3B	3B	IWB11735	2.9	Xip-II xylanase inhibitor	
	QGbg.mgb-5B	5B	IWB70546	3.2	-	
	QGbg.mgb-7A.1	7A	IWB74166	3.4	isoamylase	
	QGbg.mgb-7A.2	7A	IWB68797	3.2	fructan 1-exohydrolase	
*T. turgidum* L. ssp. *durum* cv Duilio × Avonlea RIL (F_2:7_)	QGbg.mgb-2A.1	2A	IWB1280	4.5	-	[62]
	QGbg.mgb-2B.1	2B	IWB30115	4.7	-	
	QGbg.mgb-2B.2	2B	IWB23783	3.8	β-glucosidase 1a	
*Aegilops biuncialis*	1	4M/6U	100022501_F_0	4.5	glutathione S-transferase 3-like	[78]
*Aegilops biuncialis*	2	5M	100013840_F_1	3.1	-	
*Aegilops biuncialis*	3	1M/1U	100079925_F_0	3.6	-	
*T.aestivum* L. (line Mv9kr1)	1	4ABD	100022501_F_0	4.5	microsomal glutathione S-transferase 3 *	
*T.aestivum* L. (line Mv9kr1)	2	5ABD	100013840_F_1	3.1	-	
*T.aestivum* L. (line Mv9kr1)	3	1ABD	100079925_F_0	3.6	putative peptide transporter *	

* Inferred from similarity of gene identified in relevant study with proteins on UniProt (https://www.uniprot.org/, accessed on 28 July 2023).

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
