# Peer review of "Genetic Approaches to Increase Arabinoxylan and β-Glucan Content in Wheat"

_plants, 2023, doi:10.3390/plants12183216_

Round 1
Reviewer 1 Report
In this manuscript (plants-2564547) entitled ‘Genetic approaches to increase arabinoxylan and β-glucan content in wheat’ submitted to Plants, Anneke Prins and Ondrej Kosik have summarized the current status of the research on genetics of arabinoxylan and β-glucan, as well as breeding approaches used to improve their quantity and quality in wheat grain. This topic is interesting, and I have some minor concerns to be addressed to improve the quality of this manuscript.
1. I miss at least one summarizing or concluding figure to show the breeding approaches- The table is not ‘handy’ enough.
2. Wheat genes related to biosynthesis of Arabinoxylan and β-glucan and their regulatory mechanisms should be discussed and summarized in the revised tables.
3. Wheat germplasms with great potentials in genetic improvement of arabinoxylan and β-glucan should be highlighted in the revised manuscript
4. Genome-wide distribution of genetic loci carrying wheat genes related to the biosynthesis of Arabinoxylan and β-glucan shpuld be summarized in a new figure in the revision. Molecular markers, SNPs, and genes associated with the biosynthesis of Arabinoxylan and β-gluca should be included in this figure.
5. To provide valuable sources for both wheat breeders and researchers to understand the genetics of arabinoxylan and β-glucan biosynthesis in wheat, key information about the genetic loci associated with the biosynthesis of Arabinoxylan and β-gluca in the A, B, and D sub-genome should be summarized as tables in the revision.
Author Response
We thank reviewer 1 for their valuable comments. We have addressed all their comments to our best knowledge and we believe these have significantly improved the manuscript.
- I miss at least one summarizing or concluding figure to show the breeding approaches- The table is not ‘handy’ enough.
We have added a scheme for breeding approaches, this is now Figure 3 in the manuscript.
- Wheat genes related to biosynthesis of Arabinoxylan and β-glucan and their regulatory mechanisms should be discussed and summarized in the revised tables.
As recommended, we have summarised the genes related to biosynthesis of AX and β-glucan. This is now Table 2 in the manuscript.
- Wheat germplasms with great potentials in genetic improvement of arabinoxylan and β-glucan should be highlighted in the revised manuscript.
We have added more details in the text as well as a table to highlight and summarise wheat germplasms with the potential for genetic improvement of wheat AX and β-glucan. This is now Table 4 in the manuscript.
- Genome-wide distribution of genetic loci carrying wheat genes related to the biosynthesis of Arabinoxylan and β-glucan shpuld be summarized in a new figure in the revision. Molecular markers, SNPs, and genes associated with the biosynthesis of Arabinoxylan and β-gluca should be included in this figure.
AND
- To provide valuable sources for both wheat breeders and researchers to understand the genetics of arabinoxylan and β-glucan biosynthesis in wheat, key information about the genetic loci associated with the biosynthesis of Arabinoxylan and β-gluca in the A, B, and D sub-genome should be summarized as tables in the revision.
We added two tables summarising identified QTLs related to AX (Table 3) and β-glucan (Table 6) content and quality. The information in these tables includes QTL IDs, chromosome position, nearest maker or interval, LOD score and candidate genes, where available. We have also added a figure showing the positions of genes involved in biosynthesis of AX (Figure 2A) and β-glucan (Figure 2B) in the bread wheat genome. We believe, together with Table 2, this is nicely completing the whole picture and main points of this review.
Reviewer 2 Report
The review manuscript to be revised and edited thoroughly for the below points
(1) The manuscript is very loosely written with duplication of information and the order of the content in each sub-heading is also need to be improved
(2) Most of the information is presented only in text, whereas the information of gene(s)/QTL need to be presented as Table for more clear information and reducing the text
(3) A suitable illustration can be provided for the breeding approaches that can be followed
(4) The conclusion part is too lengthy and it need to presented in a single paragraph in a comprehensive way
Author Response
We thank reviewer 2 for their valuable comments. We have addressed all their comments to our best knowledge and we believe these have significantly improved the manuscript.
(1) The manuscript is very loosely written with duplication of information and the order of the content in each sub-heading is also need to be improved
We worked thoroughly on the flow of the text. Also removed any duplications and added some details to make the delivery clearer.
(2) Most of the information is presented only in text, whereas the information of gene(s)/QTL need to be presented as Table for more clear information and reducing the text
We have added a table with genes involved in biosynthesis of AX and β-glucan. This is Table 2 in the manuscript. We also added two tables summarising identified QTLs related to AX (Table 3) and β-glucan (Table 6) content and quality. The information in these tables includes QTL IDs, chromosome position, nearest maker or interval, LOD score and candidate genes, where available. We have also added a figure showing the positions of genes involved in biosynthesis of AX (Figure 2A) and β-glucan (Figure 2B) in the bread wheat genome. We believe, together with Table 2, this is nicely completing the whole picture and main points of this review.
(3) A suitable illustration can be provided for the breeding approaches that can be followed
We have added a scheme for breeding approaches, this is now Figure 3 in the manuscript. We hope this can be followed and nicely complements the related text in the manuscript.
(4) The conclusion part is too lengthy and it need to presented in a single paragraph in a comprehensive way
We have shortened the conclusion part but with all the respect we do not believe the conclusion should be strictly “a single paragraph in a comprehensive way”. In this part, we conclude the necessity of dietary fibre in human nutrition and put this into the perspective of wheat as the major source of dietary fibre in the Western world. We also stress out the need of quantifying the fibre increase and its translation to measurable health benefits.
Round 2
Reviewer 2 Report
Though the authors have addressed the suggestions, the conclusion is too lengthy, not restrictive to one paragraph but not a one+ page, which is to signify only the salient points the authors wish to deliver to the readers.
Suitably modifying the conclusion is required before accepting the manuscript for publication.
Author Response
We thank reviewer 2 for their valuable comments.
Previously, we have addressed all their comments to the best of our knowledge and we believe these have already significantly improved the manuscript.
Recently, we have significantly shortened the conclusions to deliver only the noticeable points to the readers. As stated by the reviewer, we believe this is the last required suggestion by reviewer 2 and the last step to have this review published.
We attach the reviewed version of the manuscript with tracked changes
Round 3
Reviewer 2 Report
Can be accepted for publication.